# An Analysis of the Links between Smoking and BMI in Adolescents: A Moving Average Approach to Establishing the Statistical Relationship between Quantitative and Dichotomous Variables

**DOI:** 10.3390/children9020220

**Published:** 2022-02-07

**Authors:** Anatoly N. Varaksin, Ekaterina D. Konstantinova, Tatiana A. Maslakova, Yulia V. Shalaumova, Galia M. Nasybullina

**Affiliations:** 1Laboratory of Mathematical Modeling in Ecology and Medicine, Institute of Industrial Ecology, Ural Branch of the Russian Academy of Sciences, 620219 Ekaterinburg, Russia; varaksin@ecko.uran.ru (A.N.V.); t9126141139@gmail.com (T.A.M.); shalaumova@ecko.uran.ru (Y.V.S.); 2Department of Hygiene and Ecology, Ural State Medical University, 620028 Ekaterinburg, Russia; gdp43@yandex.ru

**Keywords:** adolescents, body mass index (BMI) category, smoking, dietary pattern, nutritional value of the diet, moving average technique, Russia

## Abstract

The aim of this study was to determine the effect of smoking on BMI in male adolescents and explore the relationship between smoking status and diet. Methods: A cross-sectional epidemiological study into the health and diet of adolescents was carried out based on a representative sample of 375 vocational school male students aged 16–17 in the city of Chelyabinsk (Russian Federation). The students and their parents filled out verified questionnaires on their socioeconomic status, diet, and smoking status. Students’ height and body weight were measured. A comparative analysis of diets was performed between groups of smokers and non-smokers (149 and 226 individuals, respectively), and the relationship between smoking, body mass index, and actual diet was estimated. The methods used included descriptive statistics, Student’s *t*-test, Mann–Whitney U test, comparison of proportions, and moving average. Results: Non-smoking adolescent boys tended to have excess body mass compared with smokers (19.0% and 12.1%, respectively). Smokers (adolescent boys) consumed less meat, cereals, beans, and cheeses and more sweet beverages, added sugar, coffee, and alcohol. The bulk of the smokers’ diet was composed of carbohydrates (*p* = 0.026) and, to a lesser extent, proteins (*p* = 0.006). Conclusions: Significant differences were discovered in the diet between smokers and non-smokers (among adolescent boys), and smoking was associated with several indicators of unhealthy diet patterns. This is an important conclusion for developing a future program that could additionally protect at-risk groups of adolescents.

## 1. Introduction

There are two major public health issues that have attained the scale of an epidemic in the 21st century: smoking and obesity among adolescents [1,2,3,4]. These widespread risk factors may impair health in adulthood. Smoking, especially when coupled with unhealthy diet patterns, increases the risk of heart disease, stroke, osteoporosis, some cancers, and other chronic diseases and is associated with an increased incidence of nonchronic diseases [5].

Obesity is associated, as a rule, with the consumption of excess food and low physical activity [6,7]. Among the factors leading to overweight and obesity, some researchers emphasize the role of lifestyle factors (for example, smoking) and dietary habits [2,8,9]. Several studies have confirmed a relationship between socioeconomic status (SES) and lifestyle factors in overweight and obese people [10,11,12]. It has been established that people of lower SES have poorer nutritional patterns and a less varied diet [13]. Evidence shows that the level of education is linked to healthy dietary habits [14]. Previous studies have revealed consistently higher current smoking prevalence among people with a lower educational level in countries across the European Region [15]. For instance, Nédó and Paulik [7] found that smoking was more prevalent among subjects with a medium educational level (OR = 1.66, 95% CI: 1.16–2.38) and poor financial conditions (OR = 3.13, 95% CI: 2.06–4.74) compared with highly educated or financially better-off people.

There are studies showing that smoking can modify the overweight risk, but data on the effect of smoking obtained in various studies are ambiguous. Thus, in a univariate analysis conducted in Hungary, obese individuals compared to non-obese were more likely to be non-smokers (OR = 0.65, 95% CI: 0.47–0.89) [7]. In a study carried out in Korea, smoking was found to be positively associated with central obesity [16]. In a longitudinal analysis of lifestyle habits in relation to body mass index performed in Sweden on subjects from the Stockholm Public Health Cohort 2002–2010 (*N* = 23,108), smoking cessation was associated with the onset of obesity in men (OR = 1.69, 1.15–2.50) and women (OR = 1.99, 1.39–2.85) [17].

It is well known that tobacco smoking in itself is the single most significant preventable cause of death. This was noted in a WHO fact sheet: “The tobacco epidemic is one of the biggest public health threats the world has ever faced, killing more than 8 million people a year around the world. More than 7 million of those deaths are the result of direct tobacco use while around 1.2 million are the result of non-smokers being exposed to second-hand smoke” [18]. Moreover, the WHO has reported more than 25 diseases, the course of which is worsened by smoking (particularly cardiovascular and pulmonary diseases and cancers) [19]. The findings reported by Carnevale et al. [20] show that smokers are prone to an elevated risk of atherosclerosis and cardiovascular diseases due to substances present in cigarette smoke that promote the formation of free radicals and plaques. The prevalence of smoking in the Russian Federation remains one of the highest in the world. Currently, about 65% of men and 30% of women in Russia smoke [15]. According to national studies [21], 7% of women and 40% of men in the 15–19 age group smoke. On average, they smoke 7 and 12 cigarettes a day, respectively. The smoking statistics, in numbers, are as follows: more than 3 million adolescents in Russia are smokers, including 0.5 million females and 2.5 males. The proportion of smokers among secondary vocational school students is 64% women and 75% for men, of whom 1 in 10 demonstrates obvious tobacco dependence [21]. All of the above smoking-related health risks are greater for adolescents since early onset is associated with a heavier smoking habit in adulthood in the future [22,23,24]. These studies also demonstrate that the majority of teenagers do not tend to maintain a balanced healthy diet.

Various researchers have reported that unhealthy food is often a concomitant risk factor for deviations from normal body weight among smoking adolescents [25,26,27]. The findings from the Project EAT (Eating among Teens) in the USA also show that smoking adolescents are less likely to participate in sports or consume enough calcium-rich foods and beverages [26].

Although previous studies have reported a relationship between smoking status and other behavioral health-related factors among adolescents, there is a lack of research into the complex relationships between smoking and eating patterns in adolescence.

Thus, the aims of this cross-sectional sampling epidemiological study were to determine the effect of smoking on BMI in male adolescents and to explore the relationship between smoking status and diet among them.

## 2. Materials and Methods

### 2.1. Design of the Study

This study is a fragment of a cross-sectional epidemiological study into the health and diet of adolescent students carried out at secondary vocational schools in the city of Chelyabinsk (Russian Federation) in 2017–2019.

The research protocol was approved by the Local Ethics Committee of the South Ural State Medical University.

### 2.2. Sample Population

The study involved students from four secondary vocational schools in the city of Chelyabinsk. The sample included all male students aged from 16 to 17 years from each of these schools, based on their informed consent to participate in the study, with the exception of those whose actual nutrition differed significantly due to their health or social condition: adolescent boys on clinical or dietary nutrition in connection with a chronic disease or after an acute illness within 14 days before the survey. As a result, we obtained data on 401 adolescent boys.

Since the aim was to explore the effect of smoking on BMI, we excluded from analysis 26 males who had smoked for less than two years. A two-year period was taken as the basis, taking into account the results of a prospective study into the relationships between smoking and weight change in 1697 adolescents in the UK [27]. Overall, the study involved 375 young males, including 226 students who had never smoked and 149 students with smoking experience of more than two years. All of the young males were indigenous inhabitants of the Ural region.

The sample was homogeneous according to many indicators: sex (only males), age (16–17-year-olds), family income, parents’ education, etc. Thus, the majority of the participants (84.3%) described their family’s level of income as “Same as others”. It should be noted that the income level was not found to be high. According to official statistics, the per capita income mode of the population in the Chelyabinsk region in 2019 was RUB 13,942 (around USD 200) a month [28].

In the sample, 54% of mothers and 56% of fathers had secondary vocational education. Higher education was reported by 28.8% of mothers and 19% of fathers; secondary education was reported by 13% of parents. Obviously, a vocational school is chosen for children by families in which the parents themselves have a secondary vocational education background. Accordingly, the adolescent males under study may be categorized as belonging to the same social class and income level. Their educational level is even more homogeneous since they were vocational school students in Chelyabinsk.

The homogeneity of the sample by sex, age, social status, income, and education is the strength of this study in terms of minimizing the effect of factors affecting BMI. For obvious reasons, some questions were not answered. Thus, 4.5% of the respondents failed to answer the question about their mother’s level of education and 11.7% about that of their father. For other indicators (required for the purposes of the study), the percentage of answers amounted to 100.

### 2.3. Measurements

Height and body weight were investigated anthropometrically in the medical offices of the colleges. The results of height and body weight measurements were used to calculate the Quetelet index (BMI = weight/height^2^, where weight is in kg and height in m). The sample of adolescent boys understudy was broken up into categories, characterizing BMI in accordance with the WHO Reference for children and adolescents aged 5–19 years [29] as follows: underweight (<P15), normal weight, and overweight (>P85).

### 2.4. Questionnaires

The students were asked three questions about smoking: “Do you smoke?” (to be answered yes or no); “At what age did you start to smoke?”; and “How many cigarettes a day do you smoke?” [30]. In our study, 39.7% of adolescents reported smoking. Later in the study, the young men were divided into two groups: “do not smoke and never smoked” (Smoke = 0) and “smoked for more than two years” (Smoke = 1). It is common practice to divide samples into two groups [25].

Diet patterns were studied using a food frequency questionnaire for the last 30 days with the indication of quantities (in g or mL) of various actually consumed foods; the list included bread, flour products, pasta, cereals and beans, potatoes, vegetables, fruit, beverages (e.g., juice), confectionary, oils and fats, sausage products, meat, fish, milk, Russian sour cream, sweet cream, cottage cheese, cheese, eggs, sugar, tea, coffee, alcohol, etc.—67 foods in all [31]. This method has been repeatedly validated for describing diet patterns in various groups of the population in Russia, including within the framework of international projects based on common approaches using a food frequency questionnaire [32].

### 2.5. Statistical Analysis

The groups of smokers and non-smokers were compared by descriptive statistics (mean values and standard errors for indicators with normal distribution; medians and their quartiles for indicators with distributions other than normal). Student’s *t*-test and Mann-Whitney test were employed for measuring the statistical significance of the differences between two means and medians. A Scheffe test was used in the analysis of variance for the multiple comparison procedure.

The moving average technique was applied to reduce the variance of the indicators studied and to reveal the structure of the relationship between the BMI variable and the indicators characterizing the smoking status and diet of the adolescents.

Data analysis was performed with the help of the STATISTICA version 10 software (TIBCO Software Inc., Palo Alto, CA, USA). The statistical significance of the differences was estimated at a significance value of α = 0.05.

### 2.6. Quantitative Outcome and Dichotomous Predictor

There are many statistical methods for establishing a link between BMI (quantitative indicator) and smoking (dichotomous variable). For the purposes of this article, these methods can be divided into two classes. In the first class, the quantitative variable is the outcome, and the dichotomous one is the predictor. Methods such as Student’s *t*-test, Kolmogorov, Cramér-von Mises tests, and others are used in this case [33,34]. In the second class, a quantitative variable is a predictor, and a dichotomous one is an outcome. The most well-known method for establishing relationships between such variables is logistic regression [35,36]. This work belongs to the first class of methods for quantitative Y as an outcome and dichotomous X as a predictor, especially parametric ones.

Among the statistical techniques used to establish a relationship between quantitative Y and dichotomous X, a special place is occupied by the Student’s *t*-test [33,37]. In particular, this parametric method makes it possible not only to test the hypothesis about the presence of a statistical relationship between Y and X (i.e., to answer the question: are there statistically significant differences between the mean values of Y in two groups for X?). This allows one to show the characteristics of groups compared in a manner understandable to medical practitioners, biologists, ecologists, etc.—we mean group average values of Y in groups and standard deviations [38].

The *t*-test helps compare two mean values of the quantitative variable Y in groups (strata) defined by two values of the dichotomous variable: and (the parentheses mean averaging). Statistically significant differences discovered between and point to a significant effect of X on Y (i.e., the presence of a relationship between Y and X). For instance, in this paper, Y means the human body mass index (BMI), and X indicates tobacco smoking status (X = 0—non-smoker, X = 1—smoker). It is known that smoking has an impact on various human health indices, including body mass [39,40]; therefore, there are good grounds to perform a statistical analysis of the relationship between BMI and smoking.

The Student’s *t*-test is used in many publications because this test is applicable to quantitative variables of Y, distributed according to the normal law, or, in the case of a large sample, the central limit theorem applies. If there is a considerable departure from these assumptions, it is recommended that non-parametric tests be used [34]. In this paper, the moving average method is compared with the Student’s *t*-test in order to establish a link between BMI and smoking.

Moving average methods are widely used in statistical analyses, particularly in time series analyses for smoothing out random fluctuations in the factor studied and highlighting trends.

There is a vast literature on this subject [41] and various applications of the moving average technique: Autoregression moving average [42], linearly weighted moving average [43], exponentially weighted moving average [44], etc. Moving average can also be applied to spatial data, excepting its use in the context of time-series modeling [45].

In this study, we used a simple method of constructing a moving average, as described in [33]. According to this method, the BMI index is sorted in ascending order, where the first stratum includes nw of the smallest BMI values (nw is the size of the moving average window). This is followed by the formation of a second stratum: shifting the window by a unit, etc. For each stratum, mean BMI and smoking status values are calculated, as a result of which the dichotomous variable “Smoking” (assuming two values: 0 and 1) turns into a quantitative one. Then, the relationship between X and Y can be analyzed by linear regression methods [46,47,48] and by analysis of variance [34,47].

It should be emphasized that this study represents a case of methods that are similar to the Student’s *t*-test but applied differently. In the Student’s *t*-test, the sample is divided into two strata by the dichotomous variable (smokers and non-smokers), with mean BMI analyzed in each of them. We suggest a contrary approach: the sample is broken up into strata by BMI, with the mean value of the dichotomous variable “Smoking” computed for each of them. In this work, we do not propose approaches other than stratification. Of course, we could apply methods such as regression to include other potentially relevant variables, but that would require a completely different approach. Our proposal is just another view on the *t*-test procedure in which there are only two variables: a dichotomous predictor and a quantitative outcome. The procedure for the application of the moving average and interpretation of the results is demonstrated in detail in the examples below.

Note that in this paper, the moving average is applied to independent observation data rather than time series and is used for solving an “inverse” (as compared with time series) problem: to reveal the structure of the relationship between variables, which is usually concealed in primary data due to the high variances of both variables, X and Y. The moving average process enables the variances of the variables to be reduced, making any possible relationship between Y and X more visible.

## 3. Results

### 3.1. Sample Characteristics

The BMI mean values and standard errors in the smoker and non-smoker groups are equal to $<\mathrm{BMI}(X=1)>=21.02\mathrm{kg}/m^{2}$ (SD = 2.73) kg/m^2^ and $<\mathrm{BMI}(X=0)>=21.46\mathrm{kg}/m^{2}$ (SD = 3.04) kg/m^2^. Thus, the average BMI value in the group of smokers proved to be less than that in the no-smoker group, but the differences in average value were very small and statistically insignificant (*p* = 0.156 by Student’s *t*-test). The median and interquartile range of BMI values is equal to 20.55 (19.48–21.59) kg/m^2^ for smokers and to 21.0 (19.76–22.79) kg/m^2^ for non-smokers. Like the mean values, the medians do not differ significantly (*p* = 0.187 by Mann-Whitney *U* test).

Using the WHO Reference [29], we determined borderline BMI values for underweight and overweight boys as follows: the percentage of underweight individuals (BMI < 18.5) was found to be 16.8% among smokers and 14.2% among non-smokers (*p* = 0.390 by the comparison of proportions test), while more non-smokers were overweight compared with smokers (BMI > 23.9) (19.0% and 12.1%, respectively (*p* = 0.260 by comparison of proportions) (Figure 1).

The consumption of the main foods and the distribution of their nutritional values in the groups of non-smokers and smokers are given in Table 1. The list of variables in the table is limited to those for which statistically significant differences in nutrition were discovered between smokers and non-smokers.

We found that smoking was associated with a number of unhealthy diet indicators. Smokers tended to consume more sugar and high added sugar and alcoholic beverages, while non-smokers consumed more cereals and beans, quark (cottage cheese), and cheese (the significance was estimated by a Mann-Whitney *U* test). For other foods, no significant differences were observed.

Non-smokers had a significantly higher contribution of proteins to their food calorie counts and cholesterol intake. Smokers had a higher contribution from carbohydrates, including simple sugars, to the calorie count of the diet and a higher ratio of carbohydrates to proteins by mass. There were no significant differences in the nutritional values of other foods.

The proportions of smoking and non-smoking adolescents in the groups were estimated by family income as described by the participants of the study (lower than others, same as others, and higher than others). Differences in the groups were not significant by the comparison of proportions test (*p* > 0.1). We also did not find differences in the mean values for the consumption of the main foods (see Table 1) among the three groups in relation to family income (Scheffe test).

### 3.2. The Moving Average Approach and t-Test: Links between Smoking and BMI

The comparison of median and mean BMI values between smokers and non-smokers, as described above, did not reveal any statistically significant differences between them. This is probably due to a large number of other factors besides smoking that can affect BMI. Since the groups of smokers and non-smokers were significantly different in terms of a number of indicators (see Table 1), the effect of smoking on BMI should be studied allowing for these differences, i.e., by methods of multifactorial statistics. In this study, we used a univariate analysis based on the moving average technique. As for the multifactorial analysis that we performed, it will be described in our next article. In particular, it will be shown that an analysis based on moving average enables, in some cases, detects differences where Student’s *t*-test and Mann-Whitney test fail to spot them.

Let us illustrate the application of the moving average technique, using as an example a study of the dependence of human body mass index *Y* on smoking status *X*. The Quetelet index is a continuous quantitative variable in this case. The dichotomous variable *X* is represented by the variable “Smoking”, assuming two values: *X* = 0 (non-smoker) and *X* = 1 (smoker).

We obtained two samples of values for the quantitative variable, *Y_i_*(*X* = 0) and *Y_i_*(*X* = 1). The distributions of *Y* (i.e., BMI) in the two groups were not different from a normal distribution (Kolmogorov–Smirnov test, *p* > 0.37), neither did we find any significant differences in *Y* variance between the groups (*t*-test, *p* = 0.152); hence, we had “ideal” conditions for *t*-test application. As noted above, the differences between the BMI average values were very small and statistically insignificant (*p* = 0.156), even though the number of observations was fairly large (375 adolescents).

This conclusion does not imply that smoking does not have a significant impact on BMI. In this particular case, we failed to detect the effect of smoking on BMI by using one specific method based on the *t*-test. What could be the reason for this conclusion? In environmental health studies, the values of BMI were not determined by the values of one dichotomous variable *X* (smoking) but also by a multitude of other covariates (for instance, nutrition, etc.). The *t*-test enables comparing the average values of the quantitative variable in the two samples, <*Y*(*X* = 0)> and <*Y*(*X* = 1)>, with each group containing all possible combinations of covariates. Such combinations of covariates can mask the impact on BMI of the dichotomous variable *X* (smoking) that is considered in the *t*-test. We believe this is why, in this example, the effect of smoking on BMI was not revealed by the *t*-test (Figure 2).

The moving average procedure (as described above) divides all values of BMI into a much larger number of groups/strata, with fewer observations in each stratum. Each stratum, therefore, is more homogeneous covariates-wise, and the relationship between BMI and the transformed dichotomous variable *X* (smoking) could manifest itself more explicitly.

Let us apply the moving average procedure to the above set of observations [49]. After applying the moving average procedure to the BMI variable for a window size of *n**_w_*** = 20, we obtained 356 strata (the number of strata is equal to 375 observations minus *n**_w_*** = 20 plus 1); for each stratum, we computed the average BMI and the average value of the dichotomous variable “Smoking”. Now the average value of the variable “Smoking” represents the proportion of smokers in a stratum; by multiplying the proportion of smokers by 100, we obtain the percentage of smokers. The result is shown in Figure 3.

We found that, for *n**_w_*** = 20, the percentage of smokers in the actual strata varied from 15 to 60 with a step size of 5 (the total number of strata was 10), while with the *t*-test applied, the percentage of smokers assumed only two values: 0 and 100. Figure 3 shows an explicit decrease in average BMI values as the percentage of smokers increases. The straight line in Figure 3 is a linear regression line. The correlation coefficient *r* = −0.404 between BMI and the percentage of smokers is statistically highly significant (*p* < 0.001), which means there is a significant relationship between BMI and smoking.

Figure 3 highlights the distribution of average BMI values in different groups by smoking. Thus, for the percentages of smokers equal to 15 and 20, we can see a narrow scattering of BMI values (from 24 to 25 units); for smoker percentages of 35 and 40, the BMI values were widely scattered (approximately from 16 to 29); finally, as the percentage of smokers increased to 50 and higher, BMI data scattering decreased again (from 17 to 22).

Figure 4 demonstrates the application of one-way ANOVA to the data in Figure 2, with the smoker percentage acting as a grouping variable. The average BMI values in the groups decreased: from 24.9 kg/m^2^ in the first group with a proportion of smokers of 15% to 18.8 kg/m^2^ in the last group with 60% smokers. This range of average BMI values (18.8–24.9 kg/m^2^) is considerable in terms of adolescent hygiene since it covers a wide span from the 25th to the 75th percentile of the WHO growth standards for young males aged 16.5 [29].

Figure 4 also enables one to analyze, in greater detail (compared to Figure 3), variations in average BMI values with an increase in the percentage of smokers; in particular, it highlights a steeper decrease in BMI with an increase in the percentage of smokers from 15 to 25 and from 45 to 60.

A comparison between Figure 2 and Figure 4 shows the significant relationship between BMI and smoking when using the moving average. Note that both figures have the same features: the axes plot BMI and smoking, and the circles and whiskers show average BMI values and related confidence intervals.

For estimating the statistical significance of the differences between the average BMI values in the groups, for Figure 4, we used a multiple comparison procedure, which replaces a *t*-test comparison if the number of groups to be compared is greater than two. The multiple comparison procedure (Scheffe test of analysis of variance) [34] reveals the following significant differences (*p* < 0.05) between the average BMI values of the groups in Figure 3: the first group differs from groups 7–10; the second group differs from groups 7–10, and the fourth group differs from groups 8–10. Thus, for the division of BMI values into 10 groups instead of the initial division into two groups (smokers and non-smokers), we observed (even applying a multiple comparison procedure, which increases *p*-values dramatically with an increase in the number of groups) significant differences in average BMI values between groups by smoker percentage. Thus, in contrast to the *t*-test, the moving average helps reveal a statistically significant effect of smoking on adolescent BMI.

We should emphasize that the moving average procedure may be used for identifying possible causes of BMI reduction. To this end, after dividing BMI values into strata by the moving average technique, it would be necessary to compute average values not only for BMI and smoking but also for various other factors that are likely to affect BMI. In our example, we computed moving averages for the consumption of some foods from the diet of the adolescents. One of the results is shown in Figure 5, which illustrates the well-known fact: smoking suppresses appetite and increases basic metabolism and, thus, can affect BMI [40,50]. Smoking also blunts the olfactory function, reducing the ability to sense the aroma of the food and affecting food appreciation. Moreover, food taste preferences change in smoking adolescents due to the suppressive effect of smoking on the taste buds on the tongue [51].

Figure 5 shows changes in eating behavior depending on the percentage of smokers in the group. Thus, the mean values of cheese consumption in the groups reduced from 27 g/day in the first group with a 15% proportion of smokers to 12 g/day in the last group with 60% smokers (Figure 5a). A similar pattern is seen in Figure 5b in relation to cholesterol intake: the mean values of cholesterol intake went down from 0.9 g/day in the first group with 15% smokers to 0.612 g/day in the last group with 60% smokers. An example of the changes in the dependence of taste preferences on smoker percentage in the group is shown in Figure 5c. It should be emphasized that Figure 5 describes the relationship between food consumption and smoking (i.e., the relationship between additional possible outcomes and the same dichotomous predictor “Smoking” and not the relationship between BMI and food consumption).

Figure 5 enables us to closely follow changes in the average values of some indicators characterizing the diet of adolescents with an increase in smoker percentage, which has become possible through the use of the moving average technique.

Many experts in biostatistics believe that, to gain a better understanding of relationships between variables *Y* and *X*, it is necessary to perform a graphic analysis of *Y*–*X* scatterplots [46,47]. The application of moving average methods enables relationships between variables studied to be visualized, which is important in applied areas, for instance, in environmental health studies.

## 4. Discussion

The aim of the study was to determine how smoking status influences BMI in male adolescents and to investigate whether there is a link between smoking status and diet.

Tobacco use is one of the main threats to human health that have ever arisen in the world, according to the WHO. According to domestic and foreign studies, the tobacco smoking habit usually develops in adolescence when the basic cohort of future smokers builds up [52]. By the early 21st century, Russia had become a world leader in terms of tobacco use growth among teenagers [53]. In our study, smoking was reported by 39.7% of male adolescents, which is in agreement with the ESPAD reports (European School Survey Project on Alcohol and Other Drugs). According to the latter, in 2011, the proportion of young men who tried smoking at the age of 13 and younger was 40% [30].

The smoking status may depend on the educational level of both teenagers themselves and their parents [53]. Thus, it was reported that, among university first-year male students (aged 17–18), 18.4% were regular smokers, 14.9% smoked sometimes, 3.4% had given up smoking, and 10.2% were passive smokers. In year five, the percentage of regular smokers among young males was already 25.5%; 18.4% smoked tobacco from time to time, 5.7% were passive smokers, and 7.8% had given up smoking [54]. In our study, the proportion of teenage male smokers was considerably higher, which is due, in our opinion, to their lower level of education (in comparison with students of higher educational institutions).

It is likely that behavioral risk factors such as smoking and an unhealthy diet are interrelated [55]. The dietary patterns revealed among smokers tend to reduce the likelihood of overweight and obesity while increasing the risk of harmful exposure to tobacco smoke, which is caused mainly by a relative shortage of irreplaceable nutrients required for neutralizing the components of tobacco smoke [56,57].

A previous study established that smokers tend to consume alcoholic beverages more frequently and eat less food rich in fiber, antioxidants, or phytochemicals, which are assumed to play a beneficial role in the prevention of chronic diseases [24]. In our study, we discovered that smoking teenagers consumed significantly more alcohol, beverages, and sugar than non-smokers, who tended to consume more food rich in fiber (cereals and beans) and protein (meat, quark, cheese).

We also found that teenage smokers consumed fewer micronutrients and more simple sugars (as a percentage of the total energy value) and carbohydrates (as a percentage of the total energy value). Additionally, the diet of teenage smokers featured a higher ratio of carbohydrates to proteins. This finding is consistent with the conclusions of other researchers [58].

The considerable differences in the diet between smokers and non-smokers that we discovered suggest that, besides weakening the appetite for food [20], smoking is likely to change the nutritional preferences of adolescents [51].

However, this suggestion should be interpreted with caution since the sample in the present study was small, and further studies are needed to confirm this tendency on a larger sample.

### Limitations and Strengths

One of the strengths of this study is the homogeneity of the sample in terms of sex, age, social status, income, and education level, which enabled us to estimate the effects of various factors on BMI in their pure (undistorted by covariates) form. The other strengths are a broad range of variables describing the structure and nutritional value of teenagers’ diets and the detailed evaluation of differences in diet between smokers and non-smokers performed with the use of the moving average technique.

In interpreting the findings of this study, one should also take into account some limitations. Firstly, the homogeneity of the sample by gender, age, social status, income level, and education background does not allow the findings to be extended to other groups of the population. Future studies should deal with samples including such other groups as well. Thus, we intend to repeat this study with a sample including girls studying at vocational schools. Secondly, although the focus of the study was on the relationships between smoking and BMI in adolescence, some important indicators related to the onset of smoking and to BMI development were not included and, hence, were not considered as covariates in the analysis. For instance, the study did not include indicators such as the physical activity and mental health of the adolescents. Thus, future studies should be designed around the physical activity levels and mental health status of young people. Additionally, a limitation of the study is the lack of data on passive smoking of teenagers in their families and vocational schools and data on the smoking status of the parents. This limitation could have affected the pattern of the relationships analyzed. The addition of passive smoking data would allow for the effect of the smoking status on BMI and on a diet to be explored in greater detail.

## 5. Conclusions

This study has presented additional evidence that smoking can affect BMI and actual diet in teenage boys.

Significant differences in diet between smokers and non-smokers have been discovered; moreover, smoking has been found to be associated with several indicators of an unhealthy diet.

This is an important conclusion for planning any future programs that could additionally protect “at-risk” groups of teenagers.

In the study reported, the crucial role belongs to the use of a homogeneous sample (male adolescents of the same age and social status) and the application of the moving average technique. The latter has allowed us to demonstrate statistical relationships that could not be revealed by traditional approaches.

## Figures and Tables

**Figure 1 children-09-00220-f001:**
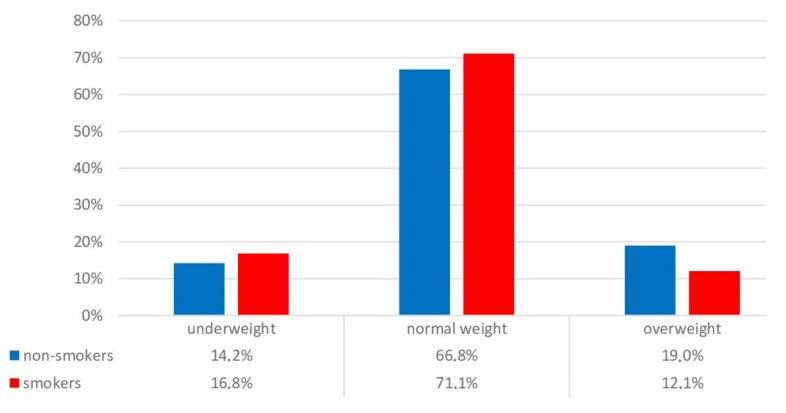
Distribution of smokers and non-smokers by categories characterizing BMI, %.

**Figure 2 children-09-00220-f002:**
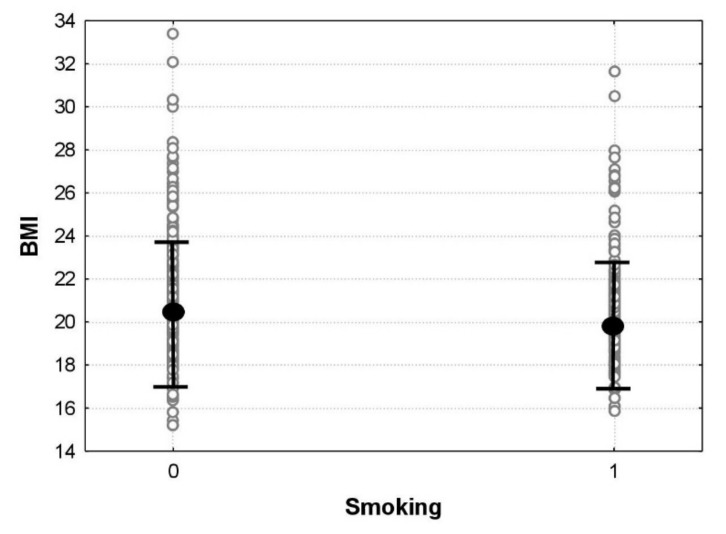
Primary data with the *t*-test applied. The smaller circles show individual BMI values for each of the 375 adolescents; the larger circles show average BMI values, and the whiskers present 95% confidence intervals for the average BMI values in the smoker and non-smoker groups.

**Figure 3 children-09-00220-f003:**
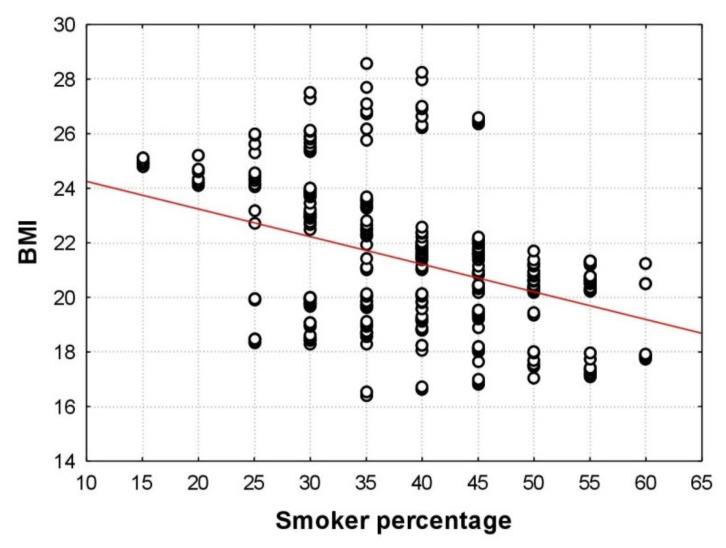
Scatterplot for the moving average data: the abscissa plots the average values of the dichotomous variable “Smoking” obtained by the moving average procedure; the ordinate shows the average BMI values in 356 strata.

**Figure 4 children-09-00220-f004:**
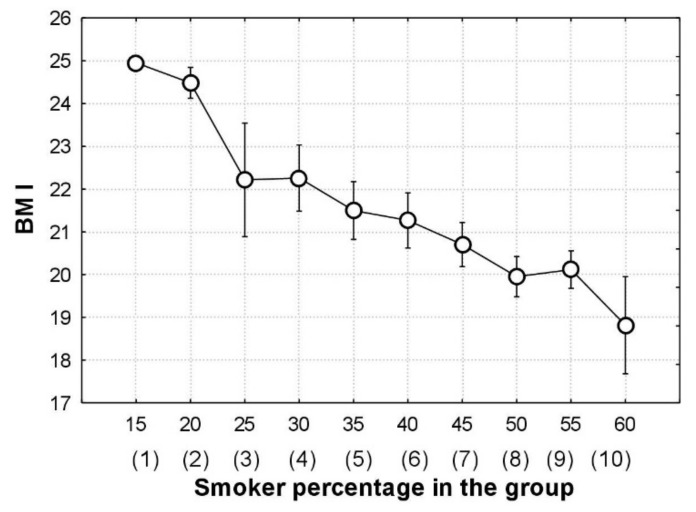
Interaction plot in one-way ANOVA; the whiskers show 95% confidence interval for the average BMI values in the groups (group numbers are shown in brackets).

**Figure 5 children-09-00220-f005:**
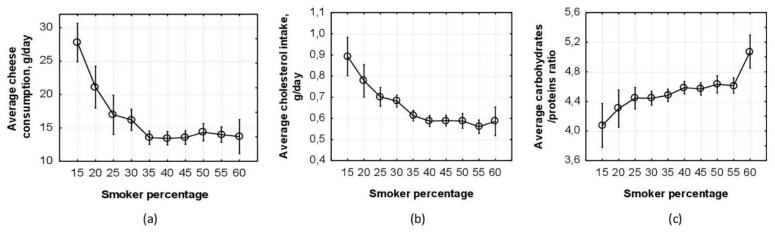
(**a**) Interaction plot in one-way ANOVA; the whiskers show a 95% confidence interval for the average values of cheese consummation in the groups with the given percentage of smokers; (**b**) interaction plot in one-way ANOVA; the whiskers show a 95% confidence interval for the average values of cholesterol intake in the groups with the given percentage of smokers; (**c**) interaction plot in one-way ANOVA; the whiskers show a 95% confidence interval for the average carbohydrates/proteins ratio in the groups with the given percentage of smokers.

**Table 1 children-09-00220-t001:** Consumption of the main foods and related nutritional values in the groups of smokers and non-smokers (P25, Median, P75).

Variable	Non-Smokers (*N*1 = 226)			Smokers (*N*2 = 149)			
	P25	Median	P75	P25	Median	P75	*p*-Value
Consumption of main foods, grams a day							
Cereals + beans	29.3	89.3	196.3	0	54.6	142.8	0.002
Beverages (juice, kompot)	46.8	94.8	214.2	71.4	142.8	285.6	0.039
Meat	80.7	148.5	234.8	59.2	119.6	183.1	0.013
Quark cheese	0	2.2	12.3	0	0	7.1	0.006
Cheese	1.0	7.1	28.5	0	4.7	21.4	0.045
Alcohol	0	0	2.5	0	1.9	8.8	<0.001
Nutritional values							
Proteins, % of energy value	10.8	12.2	13.9	10.2	11.6	13.1	0.006
Simple sugars, % of energy value	22.2	27.2	31.4	24.5	28.9	33.4	0.007
Carbohydrates, % of energy value	43.9	51.1	58.1	46.0	52.9	60.1	0.026
Cholesterol, g/day	0.3	0.6	0.9	0.3	0.5	0.8	0.022
Cholesterol/protein	3.3	4.2	5.2	3.7	4.6	5.9	0.008

## Data Availability

Data are available on request from the corresponding author.

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
