# Peer review of "An Analysis of the Links between Smoking and BMI in Adolescents: A Moving Average Approach to Establishing the Statistical Relationship between Quantitative and Dichotomous Variables"

_children, 2022, doi:10.3390/children9020220_

Round 1

Reviewer 1 Report

The authors appeared responsive to the suggestions made. I have no further suggestions at this time.

Author Response

We would like to express our appreciation to you for the thoughtful comments and helpful suggestions that helped us to improve this manuscript.

Reviewer 2 Report

This resubmission was able to address most of the previous comments. However, without investigating the relationship between diet and BMI in this group, the conclusion that "smoking can modify the effects of 
nutrition on BMI in teenagers boys" is not supported. 

Author Response

We would like to express our appreciation to you for the thoughtful comments and helpful suggestions that helped us to improve this manuscript. We have carefully considered your comments. We agree with all the comments, and we have incorporated corresponding revisions into the manuscript. Red text in the manuscript indicates changes made in response to your suggestions.

The study addresses two tasks as formulated in its title: 1) analysis of the relationship between smoking and BMI; 2) application of a novel moving-average-based approach to the first task. In the traditional approach based on the Student’s t-test, the sample is divided into two groups: smokers and non-smokers (smoking being a dichotomous variable), following which a comparison is made between the average BMI of these groups (BMI being a quantitative variable). This approach provides just one answer as to whether the average BMI values in these two groups are different or not. We propose an approach based on the moving average technique applied to the quantitative variable; as a result, the ex-dichotomous variable turns into a quantitative one: instead of two values (0-does not smoke; 1-smokes), we obtain a set of values: percentages of smokers in the strata by moving average. By applying linear regression and analysis of variance to this dataset (dataset: average BMI values and percentages of smokers in the strata), we reveal an explicit relationship between smoking and BMI: average BMI values decrease significantly as the percentage of smokers increases. On this basis, we draw a conclusion that smoking reduces BMI in 16-17 old boys covered by our study.

A question then arises, which other factors associated with smoking could affect BMI? Comparison of the groups of smokers and non-smokers by descriptive statistics in relation to various indicators enabled us to discover significant distinctions in diet between these groups. This posed another obvious question: could we reveal relationships between diet and smoking expressed as percentage of smokers? Using the same percentages of smokers as used in the examination of BMI by the moving average procedure, we discovered an explicit relationship between smoking and diet, this relationship being the same as for the relationship with BMI and detectable by linear regression and analysis of variance. To shorten the length of the paper, we omitted the results of the regression analysis since the plot of interactions in the analysis of variance demonstrates this relationship sufficiently clearly. No other objectives are pursued in the paper. In Section 4 of the paper (Discussion), we dwell upon the effect of smoking on BMI (the results our shown in Figure 1) and the effect of smoking on diet (the results are shown in Figure 2). The paper did not seek to explore the explicit relationship between BMI and diet since it should be dealt with using other methods and presents a separate challenge (which we are currently working on).

Round 2

Reviewer 2 Report

Thank you for the revision. Now the conclusion is supported by the results. 

This manuscript is a resubmission of an earlier submission. The following is a list of the peer review reports and author responses from that submission.

Round 1

Reviewer 1 Report

I have read the manuscript “An Analysis of the Links between Smoking and BMI in Adolescents: A Moving Average Approach to Establishing the Statistical Relationship between Quantitative and Dichotomous Variables” with interest. Overall the manuscript aims to determine the effect of smoking on BMI in male adolescents, and explore the relationship between smoking and diet in this population. The authors report that this is significant as there is a lack of research into complex relations between smoking and eating patterns in adolescence. Despite this, several considerations were noted which are described below.

Major considerations:

  1. Given the known associations with SES, income, and education with smoking and diet, as cited in the introduction, not having a more objective index of family income is problematic. Were differences in smoking or diet as a function of family income tested? This information should be reported.
  2. The exclusion criteria do not appear well-justified. Specifically, the rationale for excluding females who smoked, even if few in number, was unclear. As data were collected from 415 females, deciding not to analyze these data seems problematic. Please provide more information on why males and females could not be analyzed together or separately to justify this exclusion. Similarly, the reason for excluding individuals who had smoked for fewer than 2 years is unclear. If there is evidence to suggest that someone must smoke more than 2 years to have an impact on BMI, this should be cited.
  3. It is unclear what is meant by the last sentence of section 2.3. Please clarify how the WHO Child Growth Standards were used to calculate BMI.
  4. The results section indicates that “In the study, more non-smoker were found to be overweight compared with smokers (BMI > 23.9) (19.0 % and 12.1 %, respectively (r = 0.260 by comparison of proportions) (Fig. 1).” However, in the abstract, this information has been reversed, “Smoking adolescents tended to have excess body mass compared with non-smokers (19.0% and 12.1%, respectively).” Please correct this error and subsequent conclusions.
  5. The use of the “moving averages” approach does not appear warranted when more traditional ways of incorporating other potentially relevant variables using regression analyses exist. This method appears to parse the data in ways that may increase the likelihood of finding significant effects by computing a variety of averages for smoking and BMI variables as well as a “consumption of some foods from the diet of the adolescents.” Furthermore, the selection of which foods were used for these analyses is not well-characterized.
  6. The conclusions that “smoking is likely to change the nutritional preferences of adolescents” and that “smoking can modify the effects of nutrition on BMI in teenagers” are not supported by the data presented. Moreover, statements of this nature cannot be concluded using a cross-sectional sample.

Minor considerations:

  1. Citation needed for this sentence, “Tobacco kills more than 8 million people each year worldwide.”
  2. Please indicate whether compensation was provided for participation.
  3. Figure 1 should use decimal points rather than commas for percentages.

Reviewer 2 Report

The study examined a research question that is not new to the field, yet provided additional evidence about the relationships among smoking, diet quality, and obesity. The study used a new statistical method to examine a rather homogeneous population, which could benefit other studies with limited study samples. 

However, the study did a less satisfactory job determining the effect of smoking on the relationships between diet and BMI. 

  1. The results presented in the abstract (line 20-21) contradict the results in line 243-244.
  2. Several references cited did not support the statements in the Introduction. E.g., reference #16 did not report a relationship between smoking and obesity, and reference #26 was not about the Project EAT. I suggest the author take a closer look at the reference list
  3. The study did not collect information on participants' mental health, which could confound the results. This should be acknowledged in the limitation. 
  4. Although the homogeneity of the sample had its strength, obvious limitations of such should be acknowledged. 
  5. The study concluded that "smoking can modify the effects of nutrition on BMI in teenagers" (line 450), however, the study failed to examine the relationships between nutrition and BMI in its population. Thus, the conclusion was not well supported by the results.